# Diversity in the HLA-I Recognition of HLA-F Monoclonal Antibodies: HLA-F or HLA-Ib Monospecific, HLA-E or HLA-G Bispecific Antibodies with or without HLA-Ia Reactivity

**DOI:** 10.3390/antib13010008

**Published:** 2024-01-31

**Authors:** Mepur H. Ravindranath, Narendranath M. Ravindranath, Carly J. Amato-Menker, Fatiha El Hilali, Edward J. Filippone

**Affiliations:** 1Department of Hematology and Oncology, Children’s Hospital, Los Angeles, CA 90027, USA; 2Terasaki Foundation Laboratory, Santa Monica, CA 90064, USA; 3Norris Dental Science Center, Herman Ostrow School of Dentistry, University of Southern California, Los Angeles, CA 90089, USA; nravindr@usc.edu; 4Department of Microbiology, Immunology and Cell Biology, School of Medicine, West Virginia University, Morgantown, WV 26506, USA; carly.j.amato@gmail.com; 5Medico-Surgical, Biomedicine and Infectiology Research Laboratory, The Faculty of Medicine and Pharmacy of Laayoune & Agadir, Ibnou Zohr University, Agadir 80000, Morocco; f.elhilali@uiz.ac.ma; 6Division of Nephrology, Department of Medicine, Sidney Kimmel Medical College, Thomas Jefferson University, Philadelphia, PA 19145, USA; kidneys@comcast.net

**Keywords:** HLA-Ia, HLA-Ib, HLA-E, HLA-F, HLA-G, heavy chain, monomers, heterodimers, B2microglobulin, monomeric, multiplex bead assays, Luminex, mean fluorescent intensity (MFI), monospecific, polyreactive, monoclonal IgG antibodies, IgG isotypes

## Abstract

Previous investigators have used various anti-HLA-F monoclonal antibodies (mAbs) to demonstrate that the tissue distribution of HLA-F is highly restricted. Notably, these mAbs differed in their immunodiagnostic capabilities. Specifically, mAbs Fpep1.1 and FG1 detected HLA-F intracellularly in B cells but not on the cell surface, whereas mAb 3D11 detected HLA-F on the cell surface. The presence of HLA-F on T cells was recognized by mAb FG1 but not by mAb Fpep1.1. mAb 3D11 detected HLA-F on the cell surface of activated B cells and on peripheral blood lymphocytes, but not on the normal cells. Importantly, mAb 3D11 revealed that HLA-F exists as a heavy chain (HC) monomer, rather than as an HC associated with B2m. Although these mAbs are believed to be specific to HLA-F, their monospecificity has not been formally established, which is critical for immunodiagnostic and therapeutic purposes. Previously, we investigated the diversity of HLA class I reactivities of anti-HLA-E mAbs using HLA-I coated multiplex bead assays on a Luminex platform. We reported that more than 80% of the HLA-E mAbs were cross-reactive with other HLA-I molecules, with exceptionally few truly HLA-E-monospecific mAbs. In the present investigation, we generated IgG mAbs against HCs of HLA-F in Balb/C mice and examined the cross-reactivity of anti-HLA-F mAbs with other HLA-I alleles using a multiplex bead assay on the Luminex platform. Beads coated with an array of HLA homo- and heterodimers of different HLA-Ia (HLA-A, HLA-B, and HLA-C) and Ib (HLA-E, HLA-F, and HLA-G) alleles were used to examine the binding of the anti-HLA-F mAbs. Only two mAbs were HLA-F monospecific, and five were HLA-Ib restricted. Several anti-HLA-F mAbs cross-reacted with HLA-E (*n* = 4), HLA-G (*n* = 3), HLA-Ia alleles (*n* = 9), HLA-G and HLA-Ia (*n* = 2), and HLA-Ib and HLA-Ia (*n* = 6). This monospecificity and polyreactivity were corroborated by the presence of HLA-F monospecific and HLA-I-shared sequences. This study emphasizes the need to monitor the mono-specificity of HLA-F for reliable immunodiagnostics and passive immunotherapy.

## 1. Introduction

Monoclonal antibodies (mAbs) are invaluable tools for precise identification, localization, and characterization of cell surface antigens. These mAbs can be either antigen-specific or non-specific, commonly referred to as “polyreactive”. Antigen-monospecific mAbs are clinically reliable for the accurate immunodiagnosis of the antigens, especially when targeting human cancers, and for the development of passive immunotherapy aimed at antigen-bearing cancers. On the other hand, polyreactive mAbs developed against an antigen not only react with the target antigen but also with other antigens sharing identical amino acid sequences or epitopes. Therefore, it is critical to confirm the monospecificity of any mAb before utilizing it for precise identification, localization, and characterization of an antigen. This is particularly critical when dealing with protein antigens that share a common gene pool and functions, such as with human leukocyte antigens (HLAs).

The HLA class I family comprises six isotypes or isomers, namely, HLA-A, HLA-B, HLA-C, HLA-E, HLA-F, and HLA-G. Each isomer has hundreds of HLA alleles. Therefore, it is essential to rigorously examine the monospecificity or polyreactivity of the anti-HLA mAbs. For example, the mAb TFL-006, developed against HLA-E HC, has been found to react with B2m-free-HCs of almost all alleles of HLA-A, HLA-B, HLA-C, HLA-E, HLA-F, and HLA-G (based on binding to LABScreen regular, acid, and alkali-treated beadsets), but fails to bind to B2m-associated HCs (based on negative results obtained with iBeads and the LIFECODES beadset) [1,2,3]. The mAb’s binding sequence in the HLA molecules is identified as “AYDGKDY,” which remains masked in B2m-associated HLA-I but is exposed in the HCs of all HLA-A, HLA-B, HLA-C, HLA-E, HLA-F, and HLA-G alleles [1].

The polyreactivity of several HLA-specific mAbs came to light when they were tested with a Luminex multiplex beadset coated with at least 90 different HLA molecules. For example, the widely used anti-HLA-E mAb MEM-E/02 (Affinity Bioreagents, Golden, CO, USA) was employed to immunolocalize HLA-E on 19 different human cancer tissues (see Table 2 in [4]). The HLA-I reactivity of two different lots of MEM E/02 [5] was examined using dual-laser flowcytometric Luminex xMAP multiplex bead technology, and it demonstrated that MEM-EM/02 reacted with several HLA-B and HLA-C alleles in addition to HLA-E. Indeed, the binding of MEM-E/02 to the beadsets was inhibited after incubating the mAb with a most commonly shared peptide sequence containing (AYDGKDY), confirming that the mAb MEM-E/02 is a polyreactive mAb and not specifically for HLA-E [6].

Similarly, another anti-HLA-E mAb, 3D12, generated via the immunization of HLA-B27 transgenic mice with recombinant HLA-E HC purified from AEH cells (Lymphoblastoid cell line LCL 721.221 cells transfected with HLA-E gene), is also considered to be specific for HLA-E [7]. Indeed, 3D12 was also used for monitoring the cell surface distribution of HLA-E on human cancers [8]. Examining the HLA-I reactivity of mAb 3D12 using a multiplex bead assay on a Luminex platform [9], it was noted that the mAb also recognized several HLA-B and HLA-C HCs coated on the beadsets. As observed with MEM-E/02, the binding of 3D12 to HLA-E is inhibited by the peptide sequence AYDGKDY [9].

In the light of these discoveries, we conducted an examination of the HLA-I reactivity of the mAbs generated against the HCs of two HLA-E alleles in Balb/C mice. Using multiplex bead assays on a Luminex platform for HLA-I reactivity [10], we observed a wide range of specificities among the mAbs, including HLA-E monospecific; HLA-Ib monospecific; HLA-E and HLA-B bispecific; HLA-E, HLA-B, and HLA-C trispecific; HLA-E, HLA-A*1101, HLA-B, and HLA-C tetraspecific; HLA-Ia polyreactive; and HLA-Ib and HLA-Ia polyreactive [10].

In the present investigation, we generated mAbs to recombinant (deglycosylated) HLA-F HCs and studied the reactivity of the mAbs to HLA-Ia and HLA-Ib molecules using multiplex bead assays on a Luminex platform. The results reveal both monospecific and polyreactive monoclonal antibodies, which can be categorized as HLA-F or HLA-Ib monospecific, and HLA-E or HLA-G bispecific antibodies with or without HLA-Ia reactivity.

## 2. Material and Methods

### 2.1. Production of Murine mAbs against HCs of HLA-F*01:01

HLA-F HC contains 362 amino acids, including (a) the leader peptide (N-terminal 21 a.a.) cleaved in the Golgi and not present in the mature protein, and (b) a transmembrane (TM) and an intracellular (IC) domain (64 a.a) that is usually lost in soluble HLA-F HCs. The HLA-F HC used for immunization was obtained from the Immune Monitoring Laboratory, Fred Hutchinson Cancer Research Center (University of Washington, Seattle, WA, USA). The HLA-F HC is devoid of TM and IC domains (Figure 1) and is referred to as “HLA-F (FH)”. ThermoFisher One Lambda Inc (Canoga Park, Los Angeles, CA, USA) also has an HLA-F HC devoid of 26 a.a. in the extracellular domain, referred to as “HLA-F (1λ)”.

Murine monoclonal antibodies were produced following the guidelines approved by the National Research Council’s Committee on Methods of Producing Monoclonal Antibodies [10] guidelines. The mAbs generated after immunization were monitored using beads coated with HLA-F (FH), HLA-E, and HLA-G and different alleles of HLA-A, HLA-B, and HLA-C, as shown in tables.

For routine use, 50 μg of the HLA-F (FH) was diluted in 100 μL of PBS (pH 7.4) and was mixed with 100 μL of the adjuvant TiterMax (Sigma–Aldrich, St Louis, MO, USA) before injection into the footpads and peritoneum of two BALB/c mice. Since there is no animal facility at Terasaki Foundation Laboratory, the immunization of the mice was carried out at the animal facility of One Lambda Inc, Canoga Park, California. Three immunizations were given at about 12-day intervals, with an additional immunization after 12 days. Specific antibody-producing B cells were fused once with myeloma tumor cells, referred to as clones, and were cultured in a medium containing RPMI 1640 w/L-glutamine and sodium bicarbonate (Sigma–Aldrich, St. Louis, MO, USA, Cat. No. R8758), 15% fetal calf serum, 0.29 mg/mL L-glutamine/Penn-Strept (Gemini-Bio, MedSupply Partners, Atlanta, GA, USA; Cat. No. 400-110), and 1 mM sodium pyruvate (Sigma; Cat. No. S8636). Several clones were also grown using Hybridoma Fusion and Cloning Supplement (HFCS; Roche Applied Science, Indianapolis, IN, USA; Cat. No. 11363735001). No IgM Abs were detected. Almost all of the clones were cryopreserved in liquid nitrogen and supervised by Mr. Tho Pham and Mr. Vadim Jucaud at the Terasaki Research Institute (now designated as Terasaki Research Institute) in Santa Monica, CA, USA. They are still available with them for future investigations.

### 2.2. Single Antigen Beads Assay with Single-HLA Antigen-Coated Microbeads

The protocol described herein, carried out at Terasaki Foundation Laboratory, has been described earlier [10,11]. Briefly, the hybridoma culture supernatants were monitored for IgG reactivities to HLA-A, HLA-B, HLA-C, HLA-E, HLA-F, and HLA-G, by the Single Antigen Bead (SAB) assay on a Luminex platform. Mean fluorescent intensity (MFI) values were obtained for IgG antibodies reacting to HLA-coated beads of each allele of every HLA isomer. The MFI values were corrected against those obtained with negative control values for each allele.

The recombinant HLA-E, HLA-F, and HLA-G HCs (10 mg/mL in MES buffer) were obtained from the Immune Monitoring Laboratory, Fred Hutchinson (FH) Cancer Research Center (University of Washington, Seattle, WA, USA), and were custom-coated specially onto beads by the LABScreen beadset manufacturers (One Lambda Inc, Los Angeles, CA, USA) of LABScreen beadsets. Recombinant HLA-E, HLA-F, and HLA-G HCs were individually attached via a process of simple chemical coupling to 5.6-micron polystyrene microspheres, which were internally dyed with infrared fluorophores. Control beads both positive and negative were added separately and were coated with murine IgG or serum albumin (HSA/BSA), respectively. In addition, beads treated with PBS were also used as another negative control. Culture supernatants were diluted 1:50 in PBS at pH 7.2, and 20 μL was added to the 2 μL of antigen-coated microbeads (10:1). Secondary fluorescence-labeled anti-mouse polyclonal Abs were diluted 1/100 (anti-mouse IgG (H + L), Cat. No. 115-116-146, protein concentration at 0.5 mg/mL; Jackson ImmunoResearch Laboratories, West Grove, PA, USA). For identifying the isotypes, anti-Fc secondary mouse anti-isotype Abs (human-absorbed anti-mouse IgG1 (Cat. No. 1070-095) obtained from Southern Biotech, Birmingham, AL, USA) were used at 0.5 mg/mL.

Analyses were performed as previously described [10,11]. Normalized MFI of mAbs were obtained as follows: ((Trimmed MFI—MFI of mAbs treated with PBS alone)—negative control beads). The MFI cut-off used for positive HLA reactivity was 500.

## 3. Results

### 3.1. Classification of Monoclonal Antibodies Formed against HLA-F (FH)

HLA-F HCs (FH, 276 a.a.), which are devoid of leader peptide sequence (Figure 1), were used for immunization. We observed 8 distinct groups among the 32 mAbs. They were assigned unique group labels from A to H. Table 1 details the group classification and the number of mAbs developed after immunizing recombinant deglycosylated HLA-F HCs (FH). Table 1 illustrates the diverse groups (A-H) and the number of mAbs developed after immunizing recombinant deglycosylated HLA-F HCs (FH). All mAbs listed in Table 1 were positive to HLA-F but showed differential reactivity to other alleles, namely, 31 HLA-A, 50 HLA-B, 16 HLA-C, and 1 each of HLA-F and HLA-G alleles.

**Group A** (*n* = 2) represents the unique mAbs that are truly HLA-F **monospecific**. They do not recognize any alleles of other HLA-Ia or HLA-Ib alleles (MFI ranging from 0 to 200). These HLA-F monospecific mAbs may recognize amino acid sequences restricted to HLA-F.

**Group B** (*n* = 5) is unique in that the mAbs are truly HLA-Ib **monospecific**. They do not recognize any alleles of other HLA-Ia (MFI ranging from 0 to 200). These HLA-Ib monospecific mAbs may recognize amino acid sequences restricted to HLA-Ib alleles (HLA-E, -F, and -G).

**Group C** (*n* = 4) mAbs are HLA-F and HLA-E **bispecific** mAbs. They do not recognize any alleles of other HLA-Ia or HLA-G alleles (MFI ranging from 0 to 200). Evidently, these mAbs recognize amino acid sequences uniquely shared by HLA-F and HLA-E.

**Group D** (*n* = 3) mAbs are HLA-F and HLA-G **bispecific**. They do not recognize any alleles of other HLA-Ia or HLA-E alleles (MFI ranging from 0 to 200). Evidently, these mAbs recognize amino acid sequences uniquely shared by HLA-F and HLA-G.

**Group E** (*n* = 9) mAbs, while binding only to HLA-F among HLA-Ib classes, also bind to several alleles of HLA-Ia. Interestingly, they do not recognize other HLA-Ib classes, namely, HLA-E or HLA-G, suggesting that HLA-F may share sequences common to HLA-I but not found or exposed in HLA-E or HLA-G.

**Group F** (*n* = 2) mAbs recognize HLA-F and HLA-G and several alleles of other HLA-Ia; however, they do not recognize HLA-E alleles (MFI ranging from 0 to 200).

**Group G** (*n* = 1) mAb is similar to the HLA-Ib monospecific mAb but also recognizes HLA-C. The sequences shared between HLA-Ib and HLA-C need to be investigated.

**Group H** (*n* = 6) mAbs are truly HLA class I polyreactive mAbs strikingly similar to mAbs TFL-006 and TFL-007 generated after immunizing HLA-E [10]. It is well known that one of the most common amino acid sequences by almost all HLA class I molecules is ^117^AYDGKDY^123^.

### 3.2. Isotypes of IgG Monoclonal Antibodies Formed against HLA-F HC (FH)

The IgG isotype profiles of the 32 mAbs are presented in Table 2. None of them were IgG2a or IgG4. Of the 32 IgG mAbs obtained, 24 mAbs were identified to be IgG1, 6 were IgG2b, and 2 were IgG3. Anti-HLA-F MFI of 10 of 24 IgG1s mAbs were higher than that of the regular IgG used for testing. These 10 mAbs may be more sensitive to the epitopes they recognize, possibly because of their strength or concentration.

### 3.3. The HLA-F and HLA-Ib Monospecific mAbs (Groups A and B)

Table 3 shows Groups A and B monospecific mAbs. The monospecificity of Group A for HLA-F is illustrated by the mAbs HLAFS-27 and HLAFS-28 (both IgG1), which are reactive only with HLA-F coated beads but not with other HLA-Ib or HLA-Ia molecules. Similarly, the monospecificity of Group B for HLA-Ib is illustrated by the mAbs HLAFS-007 (IgG1), HLAFS-008 (IgG2b), HLAFS-019 (IgG1), HLAFS-020 (IgG1), and HLAFS-0021 (IgG2b), which are reactive with all the three HLA-Ib molecules but not with HLA-Ia molecules. Interestingly, the strength of the mAbs exemplified by MFI is higher with HLA-E than with HLA-F or HLA-G. The density of the HLA-E coated on the beads may be higher than that of other HLA-Ib loci.

### 3.4. The HLA-F and HLA-E and HLA-F and HLA-G Bispecific mAbs (Groups C and D)

Table 3 also shows Groups C and D bispecific HLA-F mAbs. The bispecificity of Group C for HLA-F and HLA-E is illustrated by the mAbs HLAFS-001 (IgG1), HLAFS-002 (IgG1), HLAFS-003 (IgG1), and HLAFS-014 (IgG2b). The strength of the mAbs exemplified by MFI is higher with HLA-E than with HLA-F. The bispecificity of Group D for HLA-F and HLA-G is illustrated by the mAbs HLA-FS-024 (IgG1), HLAFS-025 (IgG1), and HLAFS-017 (IgG2b). Although these mAbs did not show HLA-Ia reactivity in general, they showed MFI ranging from 552 to 587 for B*44:02. Since true HLA-I reactivities usually react at MFI > 1000, we deem these reactivities negative tentatively.

### 3.5. The HLA-F HC Specific mAbs Non-Reactive to HLA-E HC and HLA-G HC but Reactive to Different Alleles of HLA-Ia Loci

Figure 2 illustrates nine different HLA-F HC specific mAbs non-reactive to HLA-E HC and HLA-G HC but reactive to different alleles of HLA-Ia loci, HLA-A, HLA-B, and HLA-C (Group E), which include both B2m-free and B2m-associated HCs. The following alleles are highly reactive to all of the mAbs: **A*11:01**, A*24:02, A*24:03, A*34:01, A*36:01, B*14:01, B*15:02, B*15:11, B*18:01, B*35:01, B*37:01, B*40:01, **B*40:06**, B*44:03, B*4801, B*51:01, B*53:01, B*57:01, B*57:03, **B*58:01**, B*73:01, C*03:04, **B*05:01**, **C:08:01**, C*14:02, C*15:02 and **C*18:02**. The alleles shown in bold had the highest MFI (>10K) with HLAFS-022 and/or HLAFS-023. Possibly, the HCs of these alleles may share some common and unique sequences with HLA-F HC. Most importantly, mAb HLAFS-22 (IgG3) showed the highest MFI for all HLA-Ia reactive alleles. This is important because these mAbs do not show any reactivity with HLA-E and HLA-G.

### 3.6. The HLA-F HC Bispecific mAbs Reactive to HLA-G HC also React with Different Alleles of HLA-Ia

Table 4 documents two HLA-F HC bispecific mAbs reactive to HLA-G HC that also show positivity to different alleles of HLA-Ia loci. As was observed with HLA-F specific mAbs showing high (MFI > 5000) reactivity with different alleles of HLA-Ia, the HLA-F and HLA-G bispecific mAbs HLAFS-012 (IgG1) and HLAFS-015 (IgG1) showed MFI > 5000 for B*40:06. While mAb HLAFS-015 showed MFI > 5000 for B*47:01, HLAFS-12 remained non-reactive for the allele. HLAFS-012 also showed MFI > 5000 for C*14:02.

### 3.7. HLA-Ib Specific mAbs Reactive to HLA-Ia Alleles

Two groups fall into this category. Group G consists of HLA-Ib specific mAb reactive to HLA-C but not to HLA-A or HLA-B. Group H includes mAbs reactive to all HLA class I alleles. Table 5 shows HLAFS-006 (IgG2b) representing group G. Four HLA-C alleles showed reactivity, which include HLA-C*01:02/C*07:02/C*14:02 and C*17:01. Although the MFI values are low (<1000), four alleles showed reactivity, so it was considered as HLA-Ia reactive. Group H is illustrated in Figure 3.

### 3.8. The HLA-Ib HC Specific mAbs Reactive to Different Alleles of HLA-Ia Loci

Figure 3 shows six different HLA-F HC specific mAbs reactive to different alleles of HLA-Ia loci, HLA-A, HLA-B, and HLA-C (**Group H**), which include both B2m-free and B2m-associated HCs. HLA-I alleles reactive with high MFI (>5000) mAbs are shown in all the nine mAbs with MFI > 20,000 for C*14:02; MFI >10,000 for A*23:01, A*24:02, A*24:03, A*30:01, A*30:02, B*40:06, C*04:01, C*05:01, C*08:01, and C*18:02; MFI > 5000 for A*11:01, A*34:01, B*13:01, B*37:01, B*40:01, B*53:01, B*58:01, and C*06:02; and MFI > 4000 for C*03:04. The HCs of these alleles may share some common and unique sequences with HLA-F HC. Most importantly, mAb HLAFS-009 (IgG1), HLAFS-010 (IgG2b), and HLAFS-011 (IgG2b) showed the highest MFI for all HLA-Ia reactive alleles.

### 3.9. In Search of HLA-F and HLA-Ib Specific, HLA-F and HLA-G Bispecific and Their Unique HLA-Ia-Shared Amino Acid Sequences

Lefranc and co-investigators [12], based on the international ImMunogeneTics information system (IMGT, http://imgt.cines.fr), designated HLA-I HCs as follows: I-ALPHA chain that comprises two groove domains, the G-ALPHA 1 (D1 or α1-domain) (corresponding to the gene sequence Exon-2) and G-ALPHA 2 (D2 or α2-domain) (corresponding to Exon-3), and the C-LIKE domain (D3 or α3 domain) (corresponding to Exon-4). The Exon-4 is linked to a connecting region, a transmembrane domain (Exon-5), and intracytoplasmic domains (Exon-6, 7, and 8). One notable difference between HLA-F and other HLA class I molecules is the absence of Exon-7 and possibly Exon-8, which modifies the cell surface expression HLA-F. Therefore, the anti-HLA-F mAbs binding domains are restricted mostly to α1 and α2 domains and possibly to α3-domain of the cell surface expressed by HLA-F. However, soluble HLA-F in circulation may have all domains exposed. This may happen when we immunize utilizing the heavy chain of HLA-F in mice. Therefore, our search for HLA-F specific, HLA-F and HLA-G bispecific, and HLA-Ia-shared amino acid sequences begins with D1 or α1-domain and then extends to D2 or α2 and D3 or α3 domains.

#### 3.9.1. Similar and Dissimilar Sequences in α1-Domain of HLA-F, HLA-E, and HLA-G

Figure 4A illustrates the D1 or α1-domain of HLA-F, HLA-E, and HLA-G. The number of amino acids (a.a.) in α1-domain of HLA-G is 3+ because of the differences in the number of a.a. in the leader sequence. Figure 4B lists the identified sequences. It should be noted that although we identify the HLA-F specific, bispecific, or HLA-Ia shared sequences, the mAb may bind to linear sequences or discontinuous sequences located on the helical grooves of α1 and α2 domains. The a.a. shown in red, italic, and underlined font refers to sequences specifically for HLA-F, HLA-E, and HLA-G. These sequences occur only in HLA-F, HLA-E, or HLA-G.

This is verified by comparing the sequences of 513 HLA-A alleles, 847 HLA-B alleles, and 283 HLA-C alleles. Previously, we have shown that the specific sequence of HLA-E (^86^RSARDT^91^) was used to inhibit the binding of anti-HLA-E monospecific mAb TFL-033 to HLA-E HC coated on the beads [13]. This sequence was not observed in HLA-F, HLA-G, or any of the hundreds of HLA-I alleles examined. Peptide inhibition studies would be highly beneficial and should be considered in the future for the monospecific HLA-F mAbs. It is interesting to note that the position of the HLA-F specific sequence is located in the same region, but the sequence differs from HLA-E and HLA-G, and it is ^86^GYAKAN^91^. Figure 4B lists sequences shared by HLA-F, HLA-E, and HLA-G (amino acids shown in blue in Figure 4A). One or more of these three amino acids may be the binding site of HLA-Ib specific mAbs. Since the location is in the region of a helical configuration, the mAb may be capable of recognizing the discontinuous sequence. The amino acids sequences recognized by bispecific anti-HLA-F mAbs (mAbs that recognize HLA-F and HLA-G) are listed in Figure 4B and shown in green in Figure 4A. Although it is reasonable to infer that these sequences are a binding domain of bispecific HLA-F mAbs, peptide inhibition studies using the synthetic peptides against the binding of the bispecific mAbs to HLA-F and HLA-G are needed for confirmation.

#### 3.9.2. Similar and Dissimilar Sequences in α2-Domain of HLA-F, HLA-E, and HLA-G

Figure 5A illustrates the D2 or α2-domain of HLA-F, HLA-E, and HLA-G. Figure 5B lists the identified sequences. The a.a. shown in red, italic, and underlined font refers to sequences specific for HLA-F, HLA-E, and HLA-G. These sequences occur only in HLA-F, HLA-E, or HLA-G, and this was verified by comparing the sequences of 513 HLA-A alleles, 847 HLA-B alleles, and 283 HLA-C alleles. Previously, we showed that the specific sequence of HLA-E (^164^SEQKSNDASE^173^) was used to inhibit the binding of anti-HLA-E monospecific mAb TFL-033 to HLA-E HC coated on the beads [13]. Figure 5A also shows another sequence specifically for HLA-E. In HLA-F, we found three specific sequences not found in other HLA-I alleles (Figure 5B). In HLA-G, we found two sequences not found in other HLA-I alleles. A peptide inhibition study of the monospecific mAb of HLA-F remains to be carried out using synthetic peptides of specific sequences. Figure 5B also lists sequences shared by HLA-F, HLA-E, and HLA-G (a.a. shown in blue in Figure 5A). Strikingly and most importantly, one of these sequences, namely, AYDGKDY, is also found in 513 HLA-A alleles, 847 HLA-B alleles, and 283 HLA-C alleles examined. These two sequences may be the binding site of HLA-Ib specific mAbs but may also be considered as the shared sequences common to all HLA class I molecules. Since the location is in the region of a helical configuration, the mAb may be capable of recognizing the discontinuous sequence. The a.a. sequence LQRADPPK recognized by bispecific anti-HLA-F mAbs (mAbs that recognize HLA-F and HLA-G) is listed in Figure 5B and shown in green (sequence extends to α3 domain) in Figure 5A. Although it is reasonable to infer that this sequence is the binding domain of bispecific HLA-F mAbs, it needs to be confirmed only after peptide inhibition studies using the synthetic peptide against the binding site of the bispecific mAbs to HLA-F and HLA-G.

#### 3.9.3. Similar and Dissimilar Sequences in α3-Domain of HLA-F, HLA-E and HLA-G

Figure 6A illustrates the D3 or α3-domain of HLA-F, HLA-E, and HLA-G. Figure 6B lists the three long sequences shared by the three loci of HLA-Ib. These three sequences comprise 52 a.a. of the total 91 amino acids of the D3 domain. In addition, there is a sequence shared by HLA-F and HLA-G (LTWQRDGE), shown in green letters in Figure 6A. Since the D3 domain is fully exposed in soluble monomers of HLA-F used for immunization, one can expect the bispecific mAb production due to this sequence. However, only peptide inhibition studies can confirm the assumption. Since we did not observe bispecific mAbs recognizing both HLA-F and HLA-E, we did not elaborate on the sequences shared by these two loci.

#### 3.9.4. Similar Sequences of α1–3 Domains of HLA-F, HLA-E and HLA-G Observed in B*40:06

When we examined the HLA-Ia reactivity of nine different HLA-F specific mAbs (Figure 2), it was noted that (1) all of the nine mAbs (HLAFS-004/005/013/022/023/029/030/031 and 032) showed consistently binding affinity for (a) HLA-A*1101, A*2402, A*34:01; (b) B*13:01, B*15:11, B*40:06, B*53:01, B*57:03, B*58:01; and (c) C*03:04, C*05:01, C*08:01, C*14:02, C*15:02, and C*1802.

Most strikingly, the density or strength of binding of all the nine mAbs, as assessed by their MFI, is the highest for B*40:06, following which C*05:01 > C*08:01 > C*18:02 showed higher reactivities. Therefore, we compared the amino acid sequences of α1-domain (D1), α2-domain (D2), and α3-domain (D3) of HLA-F with HLA-B*40:06. The results unraveled unique findings as shown in Figure 7. To clarify the similarities and differences in the amino acids, we used red (in box) for similarities and blue for differences. Almost 160 a.a. out 271 amino acids in HLA-B*40:06 were in identical positions to that of the a.a. of HLA-F. However, a glance at Figure 4A reveals the presence of such a common sequence shared by HLA-F and HLA-E. However, the sequences ^86^GYAKAN^91^ in D1 and ^117^QGNNGCD^123^ and ^164^TQREYEAEEY^173^ in D2, found only in HLA-F but not in HLA-E or HLA-G, are also not observed in HLA-B*40:06. However, several sequences found in D1, D2, and D3 of HLA-F are also observed in HLA-B*40:06. It appears that the D3 sequences of HLA-B*40:06 are strikingly similar to all HLA-Ib molecules.

## 4. Discussion

### 4.1. The Diversity of the Specificities of HLA-F Monoclonal Antibodies

This investigation revealed eight distinct groups of mAbs generated after immunizing mice with HLA-F HCs (Table 1). Fourteen mAbs were HLA-Ia non-reactive (Groups A to D, Table 3). **Group A** included two monospecific mAbs, determined by non-reactivity to HLA-E, HLA-G, or any other HLA-Ia antigens. **Group B** consisted of five HLA-Ib monospecific mAbs that reacted with HAL-E, HLA-F, and HLA-G but not with HLA-A, HLA-B, or HLA-C. In addition, two groups of HLA-F bispecific mAbs (**Groups C and D**) were evaluated, consisting of four mAbs reacting to both HLA-F and HLA-E (**Group C**) and three mAbs reacting to both HLA-F and HLA-G (**Group D**). Neither Group C nor D recognized HLA-A, B, or C alleles. The remaining anti-HLA-F mAbs were reactive to HLA-Ia alleles (Groups E to H). **Group E** (Figure 2) comprised nine HLA-F specific mAbs that were HLA-E and G non-reactive but reactive to HLA-A, HLA-B, and HLA-C. **Group F** included two bispecific (HLA-F and HLA-G) mAbs reactive with HLA-Ia (Table 4). **Groups G** and **H** also reacted with both HLA-Ia and HLA-Ib loci. **Group G** was unique in that it reacted only with HLA-C (Table 5), whereas **Group H** was polyreactive with HLA-Ia and HLA-Ib alleles (Figure 3).

To identify the possible binding domains of these mAbs, the amino acid sequences of HLA-F, HLA-E, HLA-G, and alleles of HLA-Ia were examined. Nine supposed HLA-F-specific mAbs showed cross reactivity with HLA-Ia, all of which showed the strongest or highest density of binding with B*40:06 as assessed by MFI. We compared the a.a. sequence of the different domains of the HCs of HLA-B*40:06 (Figure 7) with the sequences of the α1 (Figure 4), α2 (Figure 5), and α3 (Figure 6) domains of HLA-F, HLA-E, and HLA-G. HLA-F, HLA-E, and/or HLA-G specific sequences were present in α1 (Figure 4B) and α2 (Figure 5B) domains but not in α3 domains. One sequence consisting of six amino acids (^86^GYAKAN^91^) in the α1 domain was specifically for HLA-F. Almost at the same position, different a.a’s specifically for HLA-E and HLA-G were found. In the α2-domain, two specific sequences were observed for HLA-E and HLA-G, but three sequences for HLA-F (Figure 5B). Even so, the most shared peptide sequences were found in the α2-domain. Of these sequences, ^138/141^AYDGKDY^144/147^ was found in 513 HLA-A alleles, 847 HLA-B alleles, and 283 HLA-C alleles. The sequence ^147^LNEDLRSWTA^156^ was found in about 240 HLA-A, 220 HLA-B, and 262 HLA-C alleles, and the sequences ^158^DTAAQI^163^ was modified in HLA-A but was found in 825 HLA-B and 250 HLA-C alleles. The a.a. sequence LQRADPPK recognized by bispecific anti-HLA-F mAbs (mAbs that recognize HLA-F and HLA-G) is listed in Figure 5B and shown in green (sequence extends to α3 domain). Figure 7 shows that several sequences found in D1, D2, and D3 of HLA-F are also observed in HLA-B*40:06, which explains why the high MFIs of the HLA-Ia-reactive mAbs were also high with HLA-B*40:06. It appears that the D3 sequences of HLA-B*40:06 are strikingly similar to sequences shared by all HLA-Ib molecules.

These findings demonstrate the need for caution regarding the reliability of anti-HLA-F mAbs for the specific immunodiagnosis of HLA-F used in previous reports [14,15,16,17,18,19,20,21,22,23,24,25,26]. Unless these antibodies were verified for lack of affinity for HLA-A, HLA-B, and HLA-C alleles using Luminex multiplex single antigen beads coated with at least 100 different HLA-I alleles, they would not be reliable for immunodiagnosis of HLA-F. This concurs closely with our previous studies evaluating the HLA-Ia reactivity spectrum of the so-called “HLA-E specific” mAbs such as MEM-E/02/6/7/8 [6] and 3D312 [9]. Between the two commercially available beadsets used for screening anti-HLA-Ia antibodies, the regular LIFECODES beadsets may not prove useful for the purpose of verifying the HLA-Ia HC reactivity of these mAbs without an adjusted protocol, since all the alleles on the LIFECODES beads are B2m-associated HCs, in which B2m may mask the unique as well as the shared domains. However, LIFECODES beadsets can be used after treatment with trypsin or alkali to dissociate B2m from the alleles. In contrast, as we have shown earlier [6,8,9,10,11], the commercially available beadset (LABScreen) can be used for testing the HLA-Ia reactivity of the HLA-E, HLA-F, or HLA-G specific mAbs, as LABScreen beads are coated with an admixture of B2m-free and B2m-associated HCs [11]

### 4.2. Unique Characteristics of HLA-F

Ever since the discovery of the HLA-F HCs, generated by the gene HLA-5.4 in 1990 [27], the protein structure and characteristics of HLA-F have been reported to be strikingly different from other HLA class I molecules, namely, HLA-A, HLA-B, HLA-C, HLA-E, and HLA-G. To define the structure and several of the characteristics, anti-HLA-F mAbs were required. The unique HLA-F characteristics are summarized below.

The extracellular and transmembrane domains of HLA-F HC and its CD8-binding antigen-presenting loop [28] are similar to the other HLA class I molecules, but its cytoplasmic domain is shorter than the others due to the elimination of exon 7 from the HLA-F HC mRNA [27].

There are ten highly conserved amino acid residues in the antigen recognition site of the peptide-binding groove of the classical HLA class-Ia molecules. Whereas the non-classical HLA class-Ib molecules HLA-E and HLA-G retain 8 and 9 of these residues, respectively, HLA-F possesses only 5 of the 10 highly conserved amino acid residues [16]. Five of these ten residues are altered within the HLA-F groove, which has led to the prediction that HLA-F either does not present peptide, or it does so in a fundamentally divergent manner from other HLAs [27].

There are two different glycosylation patterns on the cell surface of HLA-F HCs [16]. One has high mannose and the other a complex-type of N-glycosylation, as is observed on HLA class II DRα HCs [29]. The difference in glycosylation is thought to contribute to the differential half-life of the two forms of HLA-F (the Endo-H sensitive tapasin-independent form and the tapasin-dependent, Endo-H resistant form).

### 4.3. Tissue Distribution of HLA-F: The Need for HLA-Fmonospecific mAbs

Previous studies documented that the tissue distribution of HLA-F is highly restricted compared to other HLA-Ia and Ib molecules. The findings are primarily based on three different anti-HLA-F mAbs. It should be noted that the HLA-Ia reactivity of these mAb was not tested using either HLA-I alleles or using single antigen beads coated with multiple HLA-I alleles. These are available with LABScreen (ThermoFisher One Lambda Inc., Canoga Park, Los Angeles, CA, USA) or LIFECODES (Immucor Inc., Norcross, GA, USA) beadsets, as was performed in the previous [1,6,9,10] and present investigations.

An IgG2b mAb Fpep1.1 was raised against an HLA-F synthetic peptide corresponding to the α1 domain of the HLA-F HC, and it was reported to bind only to HLA-F but not to HLA-E or HLA-G. It was noted that HLA-F was detected intracellularly “only in B cells, B cell lines, and tissues containing B cells, in particular adult tonsil and fetal liver, a major site of B cell development” ([14] p. 327). No cell surface expression of HLA-F HCs could be observed with this mAb, and it did not detect HLA-F protein in T cell lines. *The mAb Fpep1.1 was not tested for HLA-Ia cross-reactivity.*

Another anti-HLA-F mAb FG1 (IgG2b), developed by immunizing recombinant HC refolded with B2m without a peptide, detected HLA-F in a T cell line (HUT78) as well as in endothelial cells from vessels in tonsil, spleen, and thymus [15]. This mAb also could not identify the surface expression of HLA-F on these cells. *The mAb FG1 was also not tested for HLA-Ia cross-reactivity or HLA-F monospecificity.*

Another anti-HLA-F mAb 3D11 [16] confirmed the surface expression of HLA-F HC on EBV-transformed lymphoblastoid cell lines, on all B-LCL cell lines, and on a monocyte-derived cell line. The mAb 3D11 was developed using refolded HLA-F HCs. *This mAb was also not tested for HLA-Ia cross-reactivity or HLA-F monospecificity.*

Although several cell lines had HLA-F detected intracellularly, the mAb 3D11 detected only a few cells expressing HLA-F on the surface. The mAb 3D11 did not detect HLA-F HCs on the cell surface of normal B cells or on any normal peripheral blood lymphocytes but detected cell surface HLA-F only when cells were in an activated state (caused by inflammation, injury, infection, or in the presence of proinflammatory cytokines). Lee et al. [29] concluded that the HLA-F surface expression on B cell and monocyte cell lines is partially independent from tapasin and completely independent from TAP.

Lee et al. [30], using mAb 3D11, examined whether precise lymphocyte activation induce the surface expression of HLA-F in peripheral blood mononuclear cells (PBMC) obtained from Cytomegalovirus-infected (CMV+) individuals. They stimulated mature dendritic cells loaded with a CMV peptide and assayed stimulated memory T cells by staining with anti-CD25 Ab. Only the memory CD8+ T cells specifically for the CMV peptide were activated. Notably, all activated CD3+CD25+ T cells were strongly positive to mAb 3D11, while the CD3+CD25- T cells retained the HLA-F null phenotype. Based on these findings, they concluded that activation induced the cell surface expression of HLA-F. However, the mAbs binding to HLA-F HCs with versus without B2m were not compared. This is critical because HLA-Ia HCs without B2m are well known to be expressed on activated immune cells and other cells activated by cytokines, inflammation [31,32,33,34,35], and inflammatory factors such as arthritis [36,37,38,39,40,41,42].

Using mAb 3D11, Shobu et al. [17] observed the HLA-F HC expression on the cell surface of decidual extravillous trophoblasts (EVTs) with the progression of pregnancy from mid-gestation to term, during which time the levels increased. They stated that “it is possible that EVTs in the decidua are stimulated to an activated state by the maternal immune system, though the purpose of HLA-F expression is unknown” ([17] p. 29). Indeed, Ishitami et al. [43] studied the involvement of HLA-F in pregnancy using mAb 3D11, suggesting that “similar to the MICA class I protein, HLA-F is expressed as heavy chain only,” although in vitro experimental studies support the physical interaction of B2m with refolded HLA-F HCs [43]. Although several groups used mAb 3D11, none of them tested mAb for its HLA-F monospecificity.

Furthermore, it has yet to be documented conclusively that the HCs of HLA-F are expressed together with B2m. The only evidence to date is that HLA-F-bearing cells immunostain with the mAb W6/32, which has long been considered to bind to B2m-associated HCs of HLA by almost all investigators in the field of HLA. However, a critical report of Tran et al. [44] questioned this contention and showed that mAb W6/32 “actually recognizes an epitope present on isolated non-reduced α-chains of most HLA-B allelic forms”. Indeed, Martayan et al. [45] reported on the affinity of the mAb W6/32 for HLA HCs of B2m-defective Daudi cells, confirming the above contention.

More studies [46,47,48] clarified that HLA-F HCs are expressed as monomers without peptide or B2m on the cell surface during the activated state of cells. Goodridge et al. [46] observed, in concurrence with earlier investigators, that HLA-F molecules are expressed on activated cells as monomers without a peptide or B2m. Based on the immunoprecipitation experiments performed with a “panel” of so-called “HLA-F specific mAbs”, namely, 3D11, 4B4, and 4A11, Goodridge et al. [46] state that “it appears likely that more than one complex form of HLA-F is expressed, with a portion complexed with B2m, and some or all complexed with MHC-I HC” ([46] p. 6208).

Yet another new finding emerged from the work of Dulberger et al. [49]. Using the B2m-deficient Jurkat cell assay, they validated that HLA-F monomers are a ligand for KIR3D and demonstrated that HLA-F monomers and peptid-binding HLA-F have distinct NK cell-receptor-binding partners. Based on this finding, it is suggested that the peptide binding to HLA-F obstructs its ability to recognize KIR3D, or it alters the anatomy of the HLA-F peptide-binding groove. It was also shown that LIR1 binding to HLA-F can generate an additional contact site for peptide binding. The in vitro experiments confirmed that NK receptors differentiate between peptide-bound and peptide-free HLA-F. Ho et al. [50] recovered stable peptide-bound HLA-F complexes using a soluble HLA technology. Single peptides of 8 to 21 a.a in length were identified. The structure of HLA-F bound to peptides > 10 a.a. in length suggested that HLA-F restricted peptides are only C-terminally anchored, whereas the N-terminus protrudes out sustainably from the peptide binding groove. Goodridge et al. [48] also observed that the peptide-free, B2m-free HLA-F expressed on the activated cell surface was capable of hetero-dimerizing with other monomers of HLA-I molecules (such as HLA-E and HLA-C) HHLhexpressed on the activated cells. All these structural variations of HLA-F may create stearic hindrances for mAb binding and need to be considered when utilizing the anti-HLA-F specific mAbs for immunodiagnosis.

There are several reports [18,19,20,21,22,23,24,25,26] on the expression of HLA-F on cancer cells, and in the light of what has been discussed so far, there is an imminent need for anti-HLA-F monospecific mAbs such as HLAFS-27 and HLAFS-28 potentially for therapeutic purposes. Using a rabbit polyclonal anti-human HLA-F antibody (14670-1-AP) obtained from a commercial source (Proteintec Group, Chicago, IL, USA), a group of investigators immunostained cancer tissues in patients with non-small-cell lung cancer [18], esophageal squamous cell carcinoma [19], gastric cancer [20,21], hepatocellular carcinoma [22], and breast cancer [23]. Using a commercial anti-HLA-F mAb EPR6803 (Abcam, Cambridge, UK), the expression of HLA-F was investigated in nasopharyngeal carcinoma [24] and brain glioma [25]. *The monospecificities of 14670-1-AP polyclonal antibody or mAb EPR6803 for HLA-F have neither been verified nor confirmed.* Similarly, Wuerfel et al. [26] used two different commercial antibodies purported to recognize the N-terminal end of HLA-F heavy chain (Polyclonal anti-HLA F rabbit antibody, Aviva Systems Biology, San Diego, CA, USA) and the C terminal end (polyclonal anti-HLA-F rabbit antibody, Sigma Aldrich, Taufkirchen, Germany). *It is not clear whether their HLA-F specificity was verified.*

Avoiding HLA-F monoclonal antibodies, Hrbac et al. [51] detected HLA-F overexpression in Glioblastoma (GBM) by comparing the expression of 69 GBM and 21 non-tumor brain tissue samples using RT-qPCR or a high-capacity cDNA reverse transcription polymerase (ThermoFisher Scientific) assay. Whether HLA-F is expressed on the tumor cell surface as a monomer or with B2m was not clarified.

As discussed above, much of our knowledge of the characteristics of HLA-F is based on studies utilizing mAbs (mAb Fpep1.1., mAb FG1, mAb 3D11, and mAb EPR68030) and the rabbit polyclonal antibody 14670-AP. Considering the present investigation, which documents more about the diversity of the specificities of anti-HLA-F mAbs, it is crucial that the aforementioned commercial mAbs are re-examined using different techniques including HLA coated multiplex beadsets. Without the documentation of their HLA-F monospecificity, the objective of identifying the target antigen and its characteristics cannot be fulfilled.

## 5. Conclusions and Limitations

HLA-F expressed on the cell surface may play a role in different physiological and pathological conditions, such as inflammation, infections, pregnancy, autoimmune diseases, malignancy, and transplantation. The characteristics and the tissue distribution of the cell surface expression of HLA-F reported in the literature are based on mAbs or rabbit polyclonal antibodies, which are developed after immunizing HLA-F peptide sequences, refolded recombinant HCs, or refolded recombinant HCs admixed with B2m. It should be noted that the stable configuration of the HC and B2m heterodimer depends on the presence of a peptide. Such an unstable heterodimer without a peptide is bound to dissociate upon immunization. Therefore, it is necessary to determine the reactivities of the monoclonals to HLA-Ia and/or HLA-Ib alleles using a Luminex multiplex single-antigen beads assay. Such characterization of the monospecificity of anti-HLA-F mAb using Luminex technology is a critical requirement for reliable immunodiagnostics of the target antigens and for assessing the passive immunotherapeutic potential of the mAbs.

In this context, the present investigation unravels the production of (1) HLA-F monospecific mAb; (2) HLA-Ia monospecific mAb; (3) HLA-E or HLA-G bispecific HLA-F mAbs; (4) HLA-F specific mAbs reacting to several HLA-A, HLA-B and HLA-C alleles; and (5) HLA-Ia and HLA-Ib polyreactive HLA-F mAbs. An examination of the unique and shared amino acids of HLA-F reveals that the striking diversity in the mAbs’ specificity and polyreactivity are due to differences in the antigenicity and/or immunogenicity of amino acid sequences of the different domains of HLA-F HCs. Indeed, the monospecific anti-HLA-F mAbs could serve as valuable tools for specific immunodiagnosis of HLA-F in tissues in general and for the passive immunotherapy of human cancers, in particular.

A major limitation of this study is that we did not carry out peptide inhibition studies on these mAbs, as was performed for anti-HLA-E mAbs [6,9]. There are at least five different sequences specifically for HLA-F. The mAb may bind to at least one of these sequences. Theoretically, one can expect five different monospecific mAbs, each one recognizing one of the sequences. Such a study may also unravel whether the monospecific mAbs binds to the α1 domain or α2 domain. This analysis will provide more information on the structural configuration of HLA-F on the cell surface.

Finally, we wish to state that any laboratory aiming to use monospecific mAbs for the immunodiagnosis of HLA-F, HLA-E, or HLA-G can easily develop their own mAbs using the HCs in BALB/c mice based on our protocol, within six to eight weeks. The HLA-Ib HCs are available with the Immune Monitoring Laboratory, Fred Hutchinson Cancer Research Center (University of Washington, Seattle, WA, USA).

## Figures and Tables

**Figure 1 antibodies-13-00008-f001:**
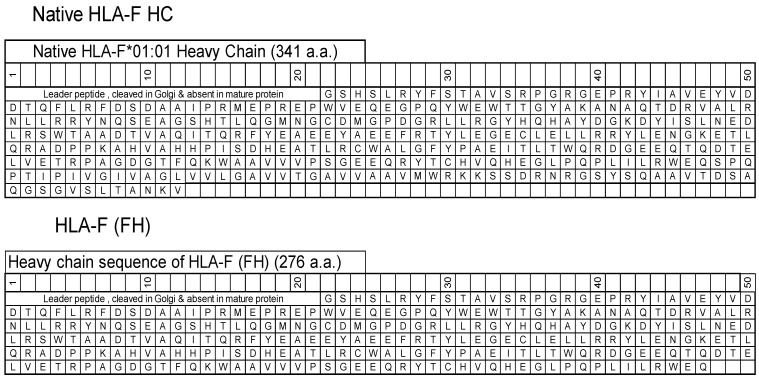
The amino acid sequences of native HLA-F*01:01 heavy chain (HC) and the HC (FH) used for immunization and for coating beads to monitor anti-HLA-F mAbs.

**Figure 2 antibodies-13-00008-f002:**
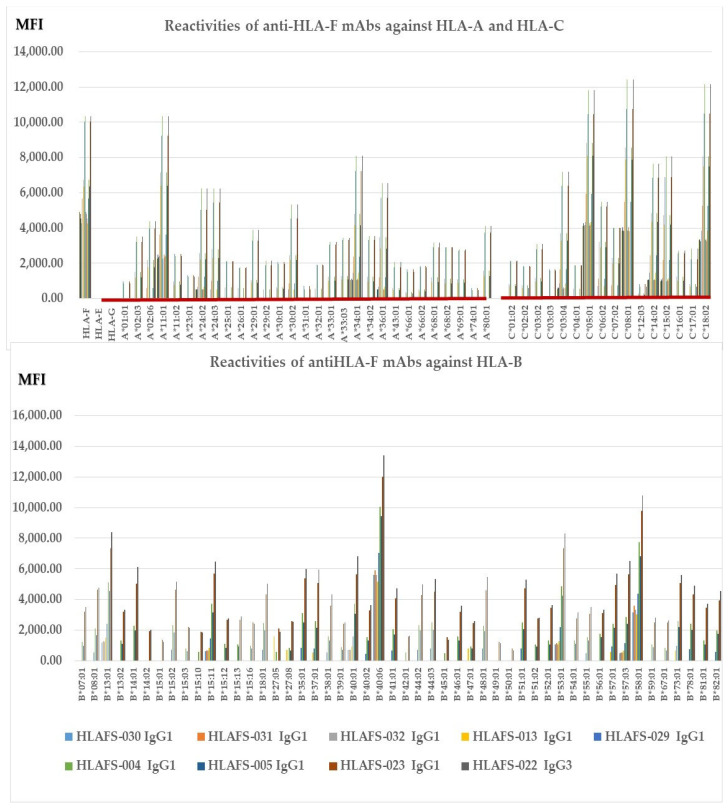
HLA-F HC specific mAbs non-reactive to HLA-E HC and HLA-G HC but reactive to different alleles of HLA-Ia loci, HLA-A, HLA-B, and HLA-C (**Group E**), which include both B2m-free and B2m-associated HCs. HLA molecules coated on One Lambda beadsets are an admixture of both B2m-associated and B2m-free HCs. Note: The following alleles are highly reactive to all of the mAbs: **A*11:01**, A*24:02, A*24:03, A*34:01, A*36:01, B*14:01, B*15:02, B*15:11, B*18:01, B*35:01, B*37:01, B*40:01, **B*40:06,** B*44:03, B*4801, B*51:01, B*53:01, B*57:01, B*57:03, **B*58:01**, B*73:01, C*03:04, **B*05:01**, **C:08:01**, C*14:02, C*15:02 and **C*18:02**. The alleles shown in bold had the highest MFI (>10K) with HLAFS-022 and/or HLAFS-023.

**Figure 3 antibodies-13-00008-f003:**
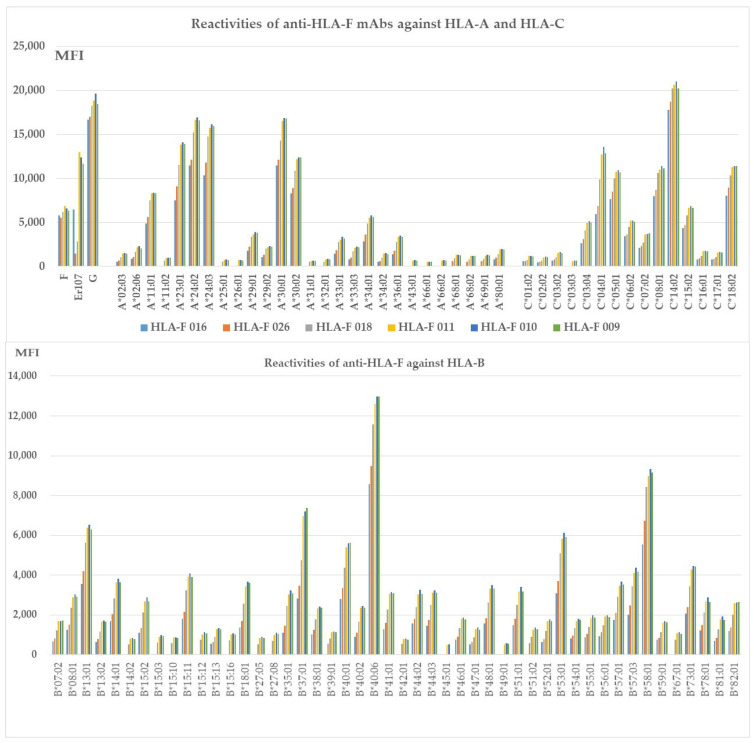
HLA-Ib specific mAbs reactive to different alleles of HLA-Ia loci, (**Group H**), which include both B2m-free and B2m-associated HCs. HLA molecules coated on One Lambda beadsets are an admixture of both B2m-associated and B2m-free HCs. *Note:* The following alleles are highly reactive to all of the mAbs: A*11:01, **A*23:01**, **A*24:02**, **A*24:03**, A*29:01, A*29:02, **A*30:01**, **A*30:02**, A*33:01, A*33:03, C*04:01, A*34:01, A*36:01, B*08:01, B*13:01 B*14:01, B*15:02, B*15:11, B*18:01, B*35:01, B*37:01, B*38:01, B*40:01, **B*40:06**, B*41:01, B*44:02, B*44:03, B*4801, B*51:01, B*53:01, B*57:01, B*57:03, B*58:01, B*73:01, B*78:01, B*82:01, **C*03:04**, **C*05:01**, C*06:02, C*07:02, **C:08:01**, **C*14:02**, C*15:02, and **C*18:02**. The alleles shown in bold had the highest MFI (> 10K) with HLAFS-022 and/or HLAFS-023.

**Figure 4 antibodies-13-00008-f004:**
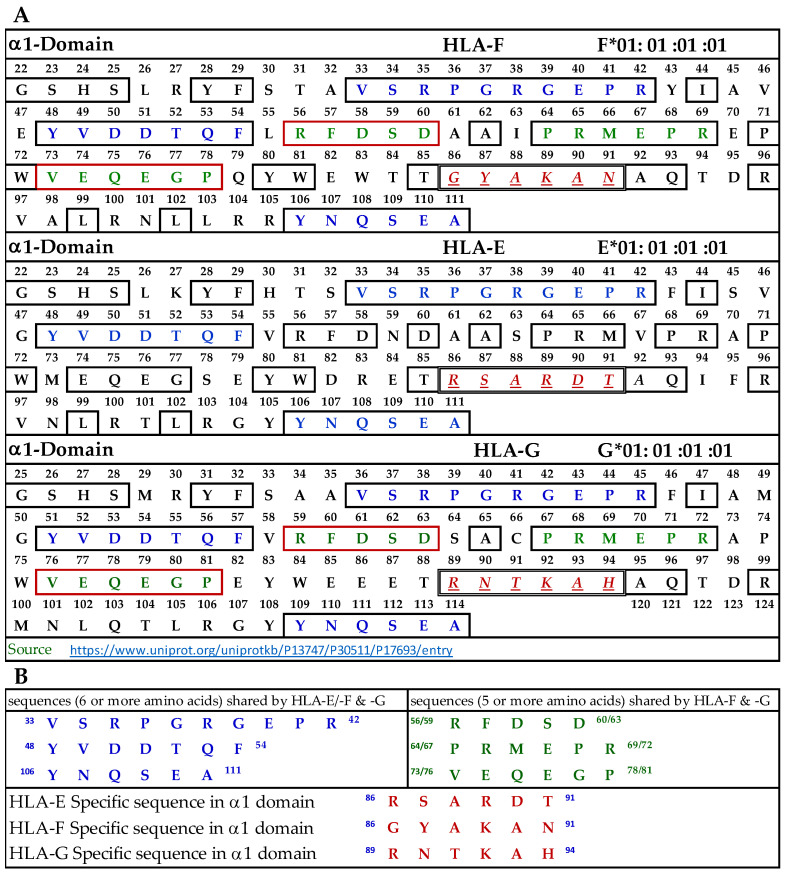
(**A**) The amino acid sequence of α1 domain of HLA-F, HLA-E, and HLA-G. (**B**) List of sequences specifically for HLA-F, HLA-Ib, and HLA-F and HLA-G.

**Figure 5 antibodies-13-00008-f005:**
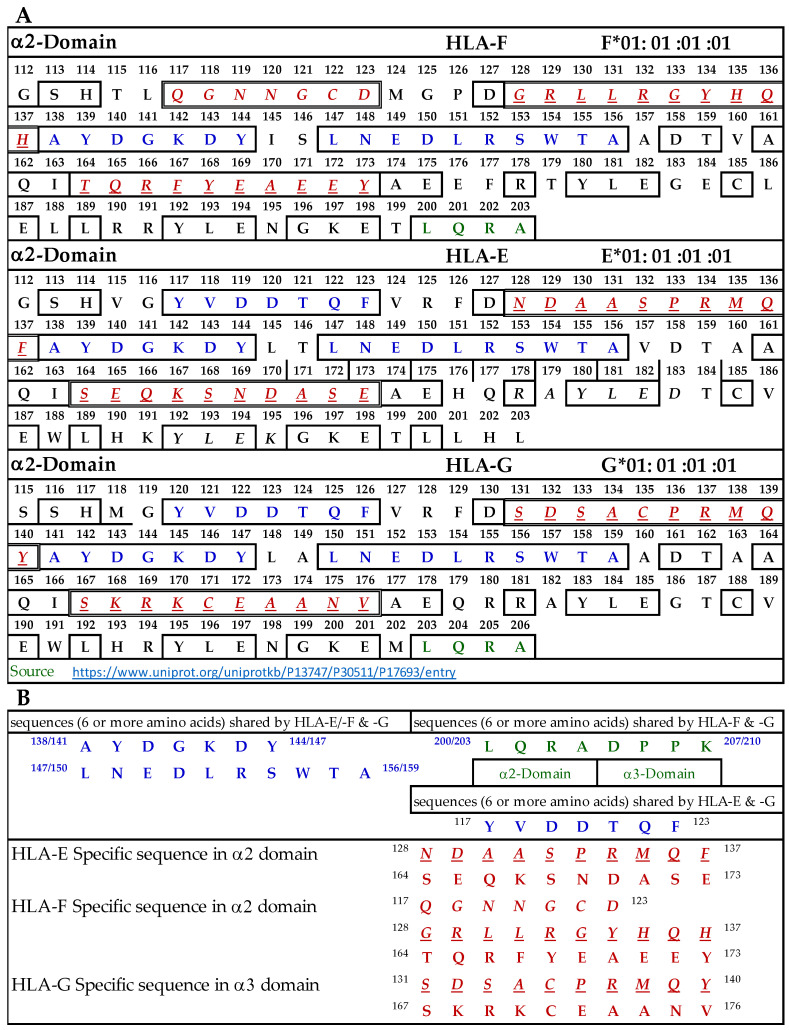
(**A**) The amino acid sequence of α2 domain of HLA-F, HLA-E, and HLA-G. (**B**) List of sequences specifically for HLA-F, HLA-Ib, and HLA-F and HLA-G.

**Figure 6 antibodies-13-00008-f006:**
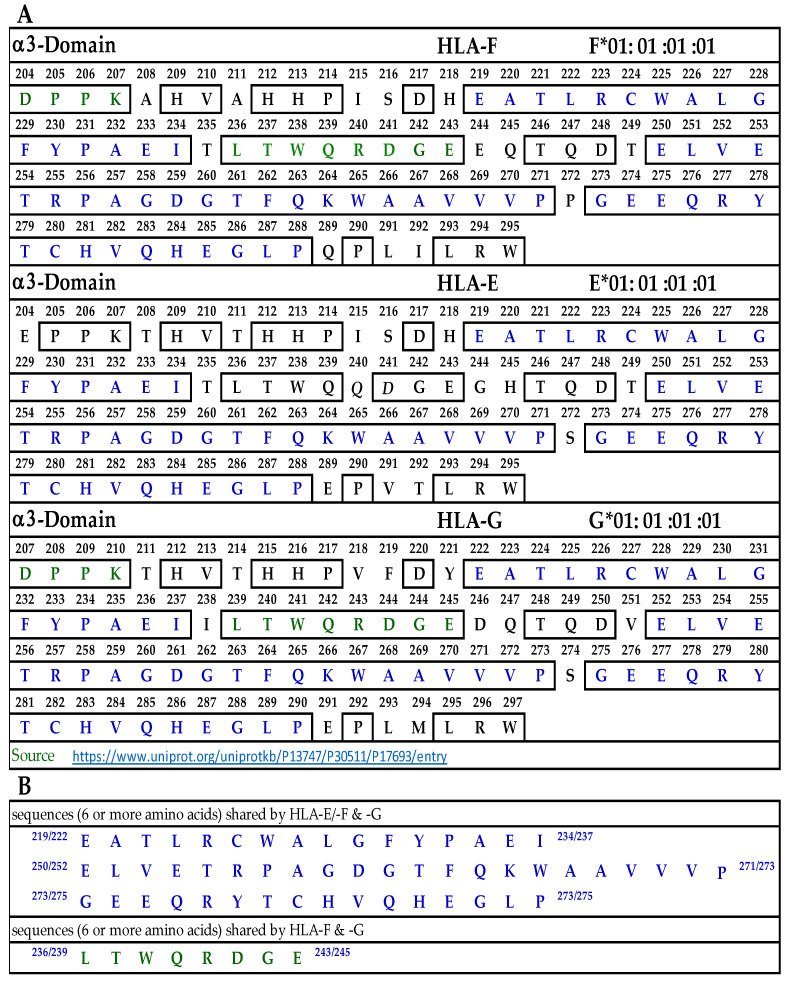
(**A**) The amino acid sequence of α3 domain of HLA-F, HLA-E, and HLA-G. (**B**) Sequences common to HLA-F, HLA-Ib and those shared by HLA-F and HLA-G.

**Figure 7 antibodies-13-00008-f007:**
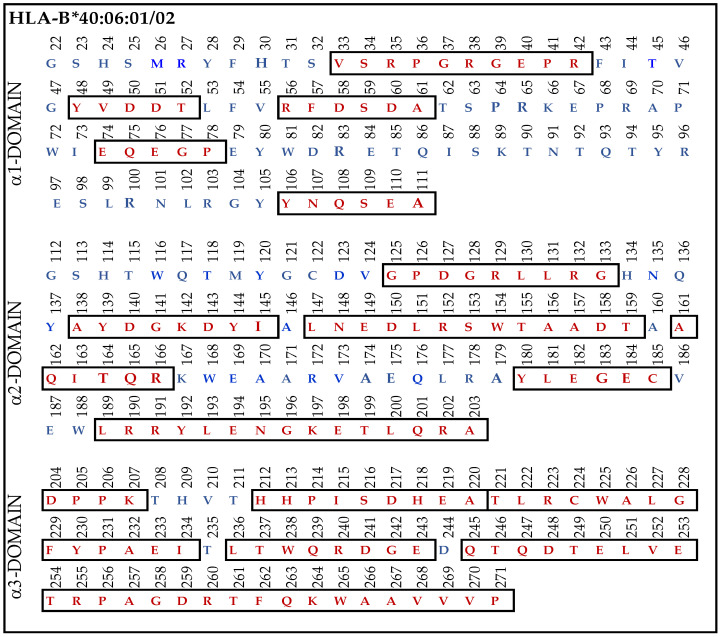
The amino acid sequence of α1, α2, and α3 domains of HLA-B*40:06:01 and B*40:06:02. The similarities (red letters in box) and differences (blue letters) in the amino acids are illustrated for each domain.

**Table 1 antibodies-13-00008-t001:** Diverse HLA class-I isoform reactivities of the monoclonal antibodies generated against recombinant non-glycosylated HLA-F HC (Source FH) are shown in bold.

Observed Group	Reactivity of Anti-HLA-F (FH) mAbs with HLA-I Alleles						Number of mAbs
	Non-Classical HLA Class-Ib			Classical HLA Class-Ia			
	HLA-F	HLA-E	HLA-G	HLA-A	HLA-B	HLA-C	
A	**Positive**	None	None	None	None	None	**2**
B	**Positive**	**Positive**	**Positive**	None	None	None	**5**
C	**Positive**	**Positive**	None	None	None	None	**4**
D	**Positive**	None	**Positive**	None	None	None	**3**
E	**Positive**	None	None	**Positive**	**Positive**	**Positive**	**9**
F	**Positive**	None	**Positive**	**Positive**	**Positive**	**Positive**	**2**
G	**Positive**	**Positive**	**Positive**	None	None	**Positive**	**1**
H	**Positive**	**Positive**	**Positive**	**Positive**	**Positive**	**Positive**	**6**

**Table 2 antibodies-13-00008-t002:** The IgG isotypes of 32 mAbs developed against HLA-F (FH) heavy chain.

Nomenclature of HLA-F mAbs	Hybridoma Supernatant ID	Isotypes of HLA-F mAbs	Anti-HLA-F MFI with	
			IgG Isotypes	Regular IgG
HLAFS-001	FT6033-1D3C9	IgG1	10,719	4099
HLAFS-002	FT6033-1D3D9	IgG1	10,200	3876
HLAFS-003	FT6033-1D3E10	IgG1	8211	4202
HLAFS-004	FT6033-2A10A3	IgG1	7843	6748
HLAFS-005	FT6033-2A10B2	IgG1	8216	6361
HLAFS-006	FT6033-3B8B5	IgG2b	2635	3856
HLAFS-007	FT6033-3B8C4	IgG1	7085	3442
HLAFS-008	FT6033-3B8E5	IgG2b	2707	3263
HLAFS-009	FT6033-4C5F2	IgG1	6363	6326
HLAFS-010	FT6033-4C5F4	IgG2b	2570	6597
HLAFS-011	FT6033-4C5H2	IgG2b	2503	6839
HLAFS-012	FT6033-5B2C1	IgG1	5758	2880
HLAFS-013	FT6033-6C6C5	IgG1	5623	4251
HLAFS-014	FT6033-6C6E3	IgG2b	1776	4040
HLAFS-015	FT6033-6C6H5	IgG1	5865	4488
HLAFS-016	FT6033-6C8B9	IgG1	5229	5791
HLAFS-017	FT6033-6C8G8	IgG2b	2111	5379
HLAFS-018	FT6033-6C8G8	IgG1	5820	6176
HLAFS-019	FT6033-7F8B10	IgG1	6211	7886
HLAFS-020	FT6033-7F8F9	IgG1	5678	8038
HLAFS-021	FT6033-7F8H11	IgG3	6382	7811
HLAFS-022	FM6034-1F2F11	IgG3	5777	10,342
HLAFS-023	FM6034-6F3G3	IgG1	4127	10,043
HLAFS-024	FM6034-7A9B8	IgG1	4164	5911
HLAFS-025	FM6034-7A9B10	IgG1	4084	5958
HLAFS-026	FM6034-7A9C7	IgG1	4012	5511
HLAFS-027	FA6035-1C1B5	IgG1	3341	3387
HLAFS-028	FA6035-1C1C2	IgG1	3123	2876
HLAFS-029	FA6035-2E1F8	IgG1	3185	5668
HLAFS-030	FA6035-9E6A7	IgG1	3077	4914
HLAFS-031	FA6035-9E6D10	IgG1	2497	4801
HLAFS-032	FA6035-9E6G8	IgG1	2742	4540

**Table 3 antibodies-13-00008-t003:** HLA-F heavy chain-monospecific (**Group A**), HLA-Ib heavy chain-monospecific (**Group B**), HLA-F HC and HLA-E HC (**Group C**) or HLA-G HC (**Group D**) bispecific mAbs non-reactive to HLA-Ia alleles, which include both B2m-free HCs and B2m-associated HCs.

Observed Groups	Lab-Codes	Anti-HLA-F Isotypes	Mean Fluorescent Intensities (MFI) Signifying the Density of mAbs on the Beads					
			HLA-F	HLA-E^R^	HLA-G	HLA-A	HLA-B	HLA-C
**Group A**	**HLA-F monospecific mAbs**							
	HLAFS-027	IgG1	3387	9	27	0	0	0
	HLAFS-028	IgG1	2876	27	26	0	0	0
**Group B**	**HLA-Ib monospecific reacting Anti-HLA-F mAbs**							
	HLAFS-019	IgG1	7886	18,197	4689	0	0	0
	HLAFS-020	IgG1	8038	19,517	5490	0	0	0
	HLAFS-021	IgG3	7811	17,831	5725	0	0	0
	HLAFS-007	IgG1	3442	9270	4813	0	0	0
	HLAFS-008	IgG2b	3263	9420	5061	0	0	0
**Group C**	**Bispecific Anti-HLA-F mAbs reacting to HLA-F and HLA-E**							
	HLAFS-001	IgG1	4099	10,151	32	0	0	0
	HLAFS-002	IgG1	3876	9735	33	0	0	0
	HLAFS-003	IgG1	4202	11,547	37	0	0	0
	HLAFS-014	IgG2b	4040	9993	42	0	0	0
**Group D**	**Bispecific Anti-HLA-F mAbs reacting to HLA-F and HLA-G**							
	HLAFS-024	IgG1	5911	7	841	0	0	0
	HLAFS-025	IgG1	59,158	10	946	0	0	0
	HLAFS-017	IgG2b	5379	8	746	0	0	0

**Table 4 antibodies-13-00008-t004:** HLA-F HC and HLA-G HC bispecific anti-HLA-F mAbs reactive with HLA-Ia alleles (**Group F**), which include both B2m-free HCs and B2m-associated HCs.

HLA Loci and Alleles	Group F. Lab-Codes and Isotypes				
	HLAFS-015 IgG1	HLAFS-012 IgG1		HLAFS-015 IgG1	HLAFS-012 IgG1
HLA-F	4488	2880	B*13:01	1742	1419
HLA-E	0	0	B*14:01	578	0
HLA-G	1529	5619	B*15:11	929	587
NC	6	6	B*18:01	500	0
PC	42	51	B*27:05	1563	0
A*11:01	2629	2077	B*37:01	809	0
A*23:01	658	826	B*40:01	1149	855
A*24:02	1908	3275	**B*40:06**	**5762**	**5021**
A*24:03	1511	2761	B*44:02	545	0
A*30:01	1092	1728	B*44:03	541	0
A*30:02	1230	1747	**B*47:01**	**8100**	0
A*34:01	1610	1020	B*48:01	551	537
A*36:01	777	407	B*51:01	512	0
			B*53:01	1458	1042
C*03:04	1072	722	B*56:01	600	0
C*05:01	4903	4096	B*57:01	695	546
C*06:02	1154	766	B*57:03	777	717
C*07:02	707	0	B*58:01	3553	2890
C*08:01	4436	4229	B*73:01	828	602
**C*14:02**	3430	**9816**	B*78:01	517	0
C*15:02	1481	1567			
C*18:02	4622	3702			

HLA-A, HLA-B and HLA-C molecules coated on 1λ beads are an admixture of both B2m-free and B2m-associated HCs [10,11].

**Table 5 antibodies-13-00008-t005:** HLA-Ib specific mAbs non-reactive to HLA-A and HLA-B but reactive to different alleles of HLA-C (**Group G**), which include both B2m-free and B2m-associated HCs.

Group G Lab-Codes and Isotypes									
HLA-Loci and Alleles	F	Er	G	A	B	C*01:02	C*07:02	C*14:02	C*17:01
HLAFS-006 IgG2b	3856	11,298	6932	0	0	691	822	514	570

## Data Availability

Data are contained within the article, and the stored mAbs producing hybridomas are at Terasaki Institute for Innovation, Los Angeles, CA, 90064, USA.

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
