# Peer review of "Diversity in the HLA-I Recognition of HLA-F Monoclonal Antibodies: HLA-F or HLA-Ib Monospecific, HLA-E or HLA-G Bispecific Antibodies with or without HLA-Ia Reactivity"

_2073-4468, 2024, doi:10.3390/antib13010008_

Round 1
Reviewer 1 Report
Comments and Suggestions for Authors
The manuscript submitted to Antibodies by Mepur H. Ravindranath and co-authors is devoted to the HLA-F Monoclonal Antibodies.
Several issues should be resolved.
1) The reviewer believes the results shown in Tables 4, 7, and similar should be located in the Supplementary. A result similar to that shown in Fig. 2 can be given once, as an example, and other similar results can be transferred in the Supplementary.
2) I would like the authors to explain in the "Discussion" section what physiological effect the observed specificity of monoclonal antibodies may have and whether it is an artifact or. On the contrary, whether it have a biological significance?
3) Line 146 - Was the site www.onelambda.com really accessed on November 4, 2011?
Sincerely
Author Response
Reply to Reviewer #1
We thank the reviewer for focusing critical attention on the results and discussion of the paper. We thank the reviewer for raising several issues, which are as follows:
- The reviewer believes the results shown in Tables 4, 7, and similar should be located in the Supplementary. A result similar to that shown in Fig. 2 can be given once, as an example, and other similar results can be transferred in the Supplementary.
We thank the reviewer for the concerns raised.
This is a paper prepared based on the model of the following previous paper published in the same journal, last year.
Ravindranath, M.H.; Ravindranath, N.M.; El Hilali, F.; Selvan, S.R.; Filippone, E.J. Ramifications of the HLA-I Allelic Reactivity of Anti-HLA-E*01:01 and Anti-HLA-E*01:03 Heavy Chain Monoclonal Antibodies in Comparison with Anti-HLA-I IgG Reactivity in Non-Alloimmunized Males, Melanoma-Vaccine Recipients, and End-Stage Renal Disease Patients. Antibodies 2022, 11, 18. Doi: /10.3390/antib 11010018
We wish to keep the same format for the tables presented in the results section so that future investigators compare the ramifications of the HLA-I allelic reactivity of anti-HLA-E mAbs with that of the diversity of the HLA-I allelic reactivity of anti-HLA-F mAbs. It is important to the primary concept of this investigation and placing it in the supplementary section will affect the focus of the readers and the objective of this paper. Since the journal has published similar extensive tables previously, presenting such tables in the Results section of the main frame of the manuscript will not be a major issue.
Coming to Figure 2 and the following figures are part of one structure of HLA-Ib molecules. Figure 2 focuses on a1-domain, whereas Figures 3 and 4 focus on a2 and a3 domains. Figure 4 focuses on all the three domains of an HLA-I molecule. Placing Figure 2 alone in the text and the rest in the supplementary section is therefore not justified.
- I would like the authors to explain in the "Discussion" section what physiological effect the observed specificity of monoclonal antibodies may have and whether it is an artifact or. On the contrary, whether it has a biological significance?
We appreciate these thought-provoking comments and the concern about whether the diversity of mAbs an artifact. We have clarified this aspect more elaborately in the Discussion. To clarify the reviewer's concern, we reply as follows:
Before we understand the specific physiological or immunological functions of cell surface HLA-F, there is a need to document its presence on the cell surface with a reliable monoclonal anti-HLA-F antibody. The very objective of this investigation is to clarify that “not all anti-HLA-F mAbs could be monospecific of HLA-F and it could polyreactive as indicated in the title. What will the polyreactive mAb serve in specific immunodiagnosis of cell surface HLA-F? It should be borne in mind that the HLA-F located on the activated cell surface can be expressed as monomeric HCs, homodimeric HCs, or even heteromeric HCs. In this regard, we wish to draw the attention of the reviewer to our recent paper entitled “Cell Surface B2m-Free Human Leukocyte Antigen (HLA) Monomers and Dimers: Are They Neo-HLA Class and Proto-HLA? Biomolecules, 2023 Jul 28;13(8):1178. doi: 10.3390/biom13081178”, in which we have clarified how the HLA configuration gets modified when cell or activated, due inflammation, or infection, or injury or malignancy. Unless, we know that the identity of the expressed HLA is confirmed, we cannot understand their physiological or immunological roles. This is true for all HLA antigens, including HLA-F, which is well known to be expressed as monomers. This imposes the need to confirm the monospecificity of HLA-F with the best available tool, the LABScreen Luminex multiplex beads coated with 90 different HLA both in intact form as well as monomers.
- Line 146 - Was the site www.onelambda.com really accessed on November 4, 2011
The error is corrected, as this line was copied from a previous publication.
Once again, we wish to thank the reviewer for his valuable comments.
Reviewer 2 Report
Comments and Suggestions for Authors
The paper is well written and presented,this is a very dense paper, at times hard to read and follow because of the nature of the subject.
The quality and soundness of the paper is supported by the experience of the authors. The paper entails a lot of fine and precise work. And it affords novel information of significance in the field.
The Discussion is mainly focused on the nature of the HLA epitopes and the reactivity characteristics of the mouse monoclonal antibodies (mAbs) generated against different HLA molecules. According to this, the fine specificity of all those mAbs is discussed. The authors point to the need of “becoming crucial that the aforementioned commercial mAbs have to be reexamined by different techniques…“ As mAbs, their VH and VL sequences may be perfectly known; so, another way of discussing their fine specifity relies on the comparison of their VH and VL, and CDRs sequences, and of their affinities.
Author Response
Reply to Reviewer # 2
We profusely thank the reviewer for considering the paper
- is well written and presented;
- is very dense due to the nature of the subject;
- quality and soundness are supported by the experience of the authors;
- entails a lot of fine and precise work;
- affords novel information of significance in the field;
Finally recognizing the need to determine the fine specificity of the mAb relies on the comparison of their VH, VL, and CDR sequences and their affinities. We will put more emphasis on this point in the revision.
Reviewer 3 Report
Comments and Suggestions for Authors
An extensive set of monoclonal antibodies has been generated and characterized within this study. However, the fact that none of these antibodies were made available in any form heavily limits the impact of this research. It would be highly valuable to the field if the best-performing antibodies were made accessible to other researchers, either by making their sequences available for recombinant expression or by commercializing them. The authors acknowledge the significant issue of cross-reactivity with currently available HLA-F antibodies, but they seem to hesitate to address this problem using the tools they have developed. Instead, they suggest that every interested laboratory should "develop their own mAbs using the HCs in BALB/c mice based on their protocol," which appears to be a wasteful use of resources and unnecessary animal experimentation.
I am also uncertain why a peptide inhibition study was not conducted to map the epitopes of at least selected antibodies, considering that the authors have the methodology in place. This would add substantial value by providing insights into the binding sites.
The presentation of the data could be more reader-friendly. I am convinced that at least some of the summary data could be better conveyed through graphical representation, while certain tables could be included in the supplementary materials.
In summary, the study is scientifically sound, but its significance is hampered by the fact that a vast library of monoclonal antibodies appears to have been generated solely for the validation of the Luminex-based cross-reactivity assay. The research would significantly increase in value if the authors had aimed to make the best-performing antibodies accessible to others.
Comments on the Quality of English LanguageThe manuscript is challenging to comprehend and would greatly benefit from substantial language editing.
Author Response
Reply to Reviewer # 3
First of all, we want to thank the reviewer for stating that “the study is scientifically sound”. We also thank the reviewer for critically reading the paper and offering valuable comments that enable our revision. Our reply to reviewer comments are as follows:
Comment 1: An extensive set of monoclonal antibodies has been generated and characterized within this study. However, the fact that none of these antibodies were made available in any form heavily limits the impact of this research. It would be highly valuable to the field if the best-performing antibodies were made accessible to other researchers, either by making their sequences available for recombinant expression or by commercializing them.
We wish to bring to the attention of the reviewer that the mAbs are available for other investigators as stated in the section Data and mAbs Availability Statement as follows:
The cryopreserved hybridomas are stored at the original location of TFL at Santa Monica. Two Research scientists were in charge of the cryopreserved hybridoma, Mr. Tho Pham and Dr. Vadim Jucaud, who carried out Luminex multiplex Immunoassay on the mAbs at TFL. A copy of the original data is available at TFL (Dr. V. Jucaud, located at Terasaki Institute for Biomedical Innovation, Los Angeles, California, 90064, USA, vjucaud@terasaki.org).
The authors acknowledge the significant issue of cross-reactivity with currently available HLA-F antibodies, but they seem to hesitate to address this problem using the tools they have developed. Instead, they suggest that every interested laboratory should "develop their own mAbs using the HCs in BALB/c mice based on their protocol," which appears to be a wasteful use of resources and unnecessary animal experimentation.
The reviewer assumes that the mAbs used by other investigators were characterized using the tools they have developed. Basically, they raised the antibody using the HLA-F recombinant Heavy chain and most of them were used on sections or cells using a fluorescinated second anti-IgG antibody. Since they used HLA-F HC they are confident that the mAbs are monospecific of HLA-F. Basically, they concluded based on the positive staining that the staining is due to their mAb. We wish to draw attention to the following paragraphs in the Discussion section. We have now highlighted the issue in red and in italics as follows
An IgG2b mAb Fpep1.1 was raised against an HLA-F synthetic peptide corresponding to the a1 domain of the HLA-F HC, and was reported to bind only to HLA-F, but not to HLA-E or HLA-G. It was noted that HLA-F was detected intracellularly “only in B cells, B cell lines, and tissues containing B cells, in particular adult tonsil and fetal liver, a major site of B cell development” [14, p.327]. No cell surface expression of HLA-F HCs could be observed with this mAb, and it did not detect HLA-F protein in T cell lines. The mAb Fpep1.1 was not tested for HLA-Ia cross-reactivity.
Another anti-HLA-F mAb FG1(IgG2b), developed by immunizing recombinant HC refolded with B2m without peptide, detected HLA-F in a T cell line (HUT78) as well as in endothelial cells from vessels in the tonsil, spleen, and thymus [15]. This mAb also could not identify the surface expression of HLA-F on these cells. The mAb FG1 was also not tested for HLA-Ia cross-reactivity or HLA-F monospecificity.
Another anti-HLA-F mAb 3D11 [16] confirmed the surface expression of HLA-F HC on EBV-transformed lymphoblastoid cell lines, on all B-LCL cell lines, and on a monocyte-derived cell line. The mAb 3D11 was developed using refolded HLA-F HCs. This mAb was also not tested for HLA-Ia cross-reactivity or HLA-F monospecificity.
…….
Using a rabbit polyclonal anti-human HLA-F antibody (14670-1-AP) obtained from a commercial source (Proteintec Group, Chicago, IL. USA), a group of investigators immunostained cancer tissues in patients with non-small-cell lung cancer [51], esophageal squamous cell carcinoma [53], gastric cancer [53, 54], hepatocellular carcinoma [55] and breast cancer [56]. Using a commercial anti-HLA-F mAb EPR6803 (Abcam, Cambridge, UK), the expression of HLA-F was investigated in nasopharyngeal carcinoma [57] and brain glioma [58]. The monospecificities of 14670-1-AP polyclonal antibody or mAb EPR6803 for HLA-F have neither been verified nor confirmed. Similarly, Wuerfel et al. [59] used two different commercial antibodies purported to recognize the N-terminal end of the HLA-F heavy chain (Polyclonal anti-HLA F rabbit antibody, Aviva Systems Biology, San Diego, CA, USA) and the C terminal end (polyclonal anti-HLA-F rabbit antibody, Sigma Aldrich Taufkirchen, Germany). It is not clarified whether their HLA-F specificity was verified.
Avoiding HLA-F monoclonal antibodies, Hrbac et al [60] detected HLA-F overexpression in Glioblastoma (GBM) by comparing the expression of 69 GBM and 21 non-tumor brain tissue samples using RT-qPCR or a high-capacity cDNA reverse transcription polymerase (ThermoFisher Scientific) assay. Whether HLA-F is expressed on the tumor cell surface as a monomer or with B2m was not clarified.
In order to document monospecificity is there any other method other than LABScreen Luminex Beadsets available? Yes, it is possible to document HLA-F using the gene expression technique. Other than that there is no technic available to ascertain whether a monoclonal antibody is monospecific. The Reviewer’s assumption that these authors have used an assay to determine the monospecificity of their mAb is incorrect. The first author studied carefully all the papers to find out how to ascertain the monospecificity of their mAb for HLA-F. Regretfully it is not available. This is also true of HLA many HLA-E monoclonal antibodies as shown in the Introduction and Discussion of this manuscript.
Our paper primarily addresses the cross-reactivity with currently available HLA-F Abs with Luminex Single antigen multiplex beads assay. We address the issues using the tools stated in the manuscript that are available in the USA.
For international scholars, obtaining these very mAbs from the sites indicated above may be challenging which is why we suggest that every interested laboratory should "develop their own mAbs using the HCs in BALB/c mice based on their protocol," When we meant every interested Lab implies labs in other countries particularly in developing countries, for it may be difficult to procure reliable mAbs from US. Moreover, many laboratories claim to have produced HLA-F mAbs in the US. Our concern is whether they have examined the monospecificity of their mAbs using the necessary tools.
Comment 2: I am also uncertain why a peptide inhibition study was not conducted to map the epitopes of at least selected antibodies, considering that the authors have the methodology in place. This would add substantial value by providing insights into the binding sites.
We have already stated as follows in the Conclusion section: “A major limitation of this study, is that we are yet to carry out peptide inhibition studies on these mAbs, as was done for anti-HLA-E mAbs [6, 9].”
In fact, we have started testing with the epitopes identified. Some of the specific sequences listed in Figure 2A do inhibit our monospecific mAbs. However, the study has to be completed using dosimetry (dose-dependent dilution of peptides against dose-dependant dilution of mAbs, as we have done earlier for anti-HLA-E mAbs.
Regretfully, we could not complete the study for the following reasons: 1. when the study was completed our mentor and the CEO Prof. Paul I. Terasaki, passed away; 2. the new director, a pharmacologist had different plans and ambitions, which did not encourage to continue the project. That is why we have stated that we are yet to carry out the peptide inhibition study. We agree with the reviewer that such an investigation “would add substantial value by providing insights into the binding sites.
We wish to bring to the attention of the Reviewer that peptide inhibition experiments are important to confirm the epitope specificity of the monospecific mAbs. However, proving that the mAbs are monospecific using bead sets coated with more than 90 HLA alleles is sufficient to establish monospecificity. No doubt that assessing epitope specificity is important to further characterize the monospecific mAb. First monospecificity has to be established for immunodiagnostic purposes.
Comment 3: The presentation of the data could be more reader-friendly. I am convinced that at least some of the summary data could be better conveyed through graphical representation, while certain tables could be included in the supplementary materials.
We thank the reviewer for the concern raised.
This is a paper prepared based on the model and style of the previous paper published in the same journal, last year. Therefore, we retained the same pattern of presentation as we are presenting to the same journal.
Ravindranath, M.H.; Ravindranath, N.M.; El Hilali, F.; Selvan, S.R.; Filippone, E.J. Ramifications of the HLA-I Allelic Reactivity of Anti-HLA-E*01:01 and Anti-HLA-E*01:03 Heavy Chain Monoclonal Antibodies in Comparison with Anti-HLA-I IgG Reactivity in Non-Alloimmunized Males, Melanoma-Vaccine Recipients, and End-Stage Renal Disease Patients. Antibodies 2022, 11, 18. Doi: /10.3390/antib 11010018
- We wish to keep the same format for the tables presented in the results section so that future investigators compare the ramifications of the HLA-I allelic reactivity of anti-HLA-E mAbs with that of the diversity of the HLA-I allelic reactivity of anti-HLA-F mAbs. It is important to the primary concept of this investigation and placing it in a supplementary section will affect the focus of the readers and objective of this paper. Since the journal has published similar extensive tables previously, presenting such tables in the Results section of the main frame of the manuscript will not be a major issue.
Comment 4: In summary, the study is scientifically sound, but its significance is hampered by the fact that a vast library of monoclonal antibodies appears to have been generated solely for the validation of the Luminex-based cross-reactivity assay. The research would significantly increase in value if the authors had aimed to make the best-performing antibodies accessible to others.
There is no doubt this study highlights not only the importance of the diversity of specificities of the anti-HLA-F mAbs but also the importance of Luminex single antigen Beads assay as an invaluable tool to monitor the specificity of the mAbs. There are many anti-HLA-F mAbs as shown in the discussion. Can the investigators confirm their monospecificity without Luminex technology? If the monospecificity of the monoclonal antibody cannot be confirmed what kind of diagnostic or therapeutic purpose it would serve?
Agreeing with the Reviewer’s recommendation, we have revised the manuscript.
Yes, these mAbs are accessible to others, as follows under Data and mAbs Availability Statement: The cryopreserved hybridomas are stored at the original location of TFL at Santa Monica. Two Research scientists were in charge of the cryopreserved hybridoma, Mr. Tho Pham and Dr. Vadim Jucaud, who carried out Luminex multiplex Immunoassay on the mAbs at TFL. A copy of the original data is available at TFL (Dr. V. Jucaud, located at Terasaki Institute for Biomedical Innovation, Los Angeles, California, 90064, USA, and vjucaud@terasaki.org).
Round 2
Reviewer 1 Report
Comments and Suggestions for Authors
I'm not sure the tables like Table 7 are worth to be published in the main part of the manuscript, not in supplementary, as well as figures Fig. 3-5. I suggest the Editor should take a balanced opinion on these issues.
Sincerely
Author Response
Reply to Reviewer # 1
I'm not sure the tables like Table 7 are worth to be published in the main part of the manuscript, not in supplementary, as well as figures Fig. 3-5. I suggest the Editor should take a balanced opinion on these issues.
Tables like 4 and 7 reflect the diversity or ramification of HLA-I loci polyreactivity of (1) HLA-F specific (not monospecific)[Table 4] and HLA-Ib reactive monoclonals [Table 7].
Each table has three different information; They are (1) HLA-I non-reactive (0<500 MFI) mAbs, (2) pattern of the high MFI reactivity (MFI in bold) and, (3) consistency the polyreactivity towards selected HLA-Ia alleles. This may not be specific interest for general readers but it has high significance of those working monoclonal antibodies directed against HLA-F, HLA-E and HLA-G, the non-classical HLA.
We have already showed similar table for anti-HLA-E mAbs in our previous publication in this journal
Ravindranath, M.H.; Ravindranath, N.M.; El Hilali, F.; Selvan, S.R.; Filippone, E.J. Ramifications of the HLA-I Allelic Reactivity of Anti-HLA-E*01:01 and Anti-HLA-E*01:03 Heavy Chain Monoclonal Antibodies in Comparison with Anti-HLA-I IgG Reactivity in Non-Alloimmunized Males, Melanoma-Vaccine Recipients, and End-Stage Renal Disease Patients. Antibodies 2022, 11, 18. Doi: /10.3390/antib 11010018
In future, when these HLA-Ia polyreactivities of HLA-F and HLA-E mAbs are compared, a new concept on such monoclonal antibodies formed in normal healthy sera will emerge. For this a reader must refer to another previous paper on normal sera and IVIg.
Ravindranath, M.H.; Terasaki, P.I.; Pham, T.; Jucaud, V.; Kawakita, S. Therapeutic preparations of IVIg contain naturally occurring anti-HLA-E Abs that react with HLA-Ia (HLA-A/-B/-Cw) alleles. Blood, 2013, 121, 2013–2028. Doi: 0.1182/blood-2012-08-447771.
Based on these specific background information alone (that is when HLA-Ia polyreactive anti-HLA-Ib mAbs are compared with polyreactivity of normal sera), new hypothesis is bound to emerge. There as an experienced scientist (25 yrs) on this mAb field, I strongly emphasize the need to include these tables in the manuscript as it was done recently for anti-HLA-E mAbs in the paper published in 2022 Antibodies.
I do not want to place them in supplementary files.
The reviewer has raised a similar issue for figures 3 to 5. These figures are the main essence of this manuscript. These figures are as important as the Tables 4 and 7. I am for shunting these figures to supplement files. Once again I emphasize that they should not be moved to supplement files, these figures tell a long story of the diversity and commonness of the amino acid sequences of the HLA-Ib and one HLA-Ia allele.